# Contactless Fingerprint Recognition Using Deep Learning—A Systematic Review

**A M Mahmud Chowdhury and Masudul Haider Imtiaz ***

Department of Electrical and Computer Engineering, Clarkson University, Potsdam, NY 13699, USA
* Correspondence: mimtiaz@clarkson.edu

**Abstract:** Contactless fingerprint identification systems have been introduced to address the deficiencies of contact-based fingerprint systems. A number of studies have been reported regarding contactless fingerprint processing, including classical image processing, the machine-learning pipeline, and a number of deep-learning-based algorithms. The deep-learning-based methods were reported to have higher accuracies than their counterparts. This study was thus motivated to present a systematic review of these successes and the reported limitations. Three methods were researched for this review: (i) the finger photo capture method and corresponding image sensors, (ii) the classical preprocessing method to prepare a finger image for a recognition task, and (iii) the deep-learning approach for contactless fingerprint recognition. Eight scientific articles were identified that matched all inclusion and exclusion criteria. Based on inferences from this review, we have discussed how deep learning methods could benefit the field of biometrics and the potential gaps that deep-learning approaches need to address for real-world biometric applications.

**Keywords:** biometrics; contactless fingerprint; deep learning; fingerprint analysis; fingerprint recognition



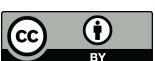

## 1. Introduction

Contactless fingerprint identification technology has the potential to be one of the most reliable techniques for biometric identification [1,2]. The first contactless fingerprint recognition system was introduced in 2004 [3] as an alternative to traditional contact-based fingerprinting [4]. Since then, interest has grown, as shown by a continually growing number of publications by different research groups. This publication corroborates that the demand for contactless fingerprint recognition systems is increasing rapidly [5]. The National Institute of Standards and Technology (NIST) has also reported that contactless fingerprint recognition system is an important component of next-generation fingerprint technologies [6]. Generally, a contactless fingerprint system involves a high-resolution camera [7,8]. The captured images provide the details of fingerprints (ridge, valleys) and wrinkles, etc. [9]. One of the challenges of the traditional contact-based fingerprint recognition system is fingerprint capturing [10]. During the acquisition of a contact-based fingerprint, issues such as a latent fingerprint left by a previous user on the sensor surface lead to low fingerprint quality [10–12]. Also, deformation and distortion of fingerprints occur because of the pressure on the sensor surface [12]. Distortions can be caused by non-uniformity of the finger pressure on the device, finger ridge changes due to heavy labor or injuries, different illumination on finger skin, or motion artifacts during image capturing [13]. When fingerprints contact the scanner, the ridge flow may become discontinuous. A lot of background noise might also be introduced during capture [14]. Often, only a partial fingerprint is obtained because the rest might be either lost or smudged during capture [8], as shown in Figure 1. This process is subject to partial information, poor quality, distortions, and variations, including background and illumination [15]. The variations in sensors and the acquisition environment may introduce a wide range of intra-

and inter-class variability in the captured fingerprint in terms of resolution, orientation, sensor noise, and skin conditions. A finger photo acquired by a contactless sensor does not suffer from deformation or latent, hidden fingerprints [7,10]. However, new challenges are also present here. For example, captured images can be of poor quality, with different size, low resolution, background segmentation, or uncontrolled illumination, and face difficulty in extracting features like minutiae, finger enhancement, etc. [16]. According to the NIST, a standard fingerprint image, generally the frontal region of the finger, requires 500 dpi imaging sensors for a good-quality application [17]. These can be captured by smartphones or a handheld electronic device [7,18–20].

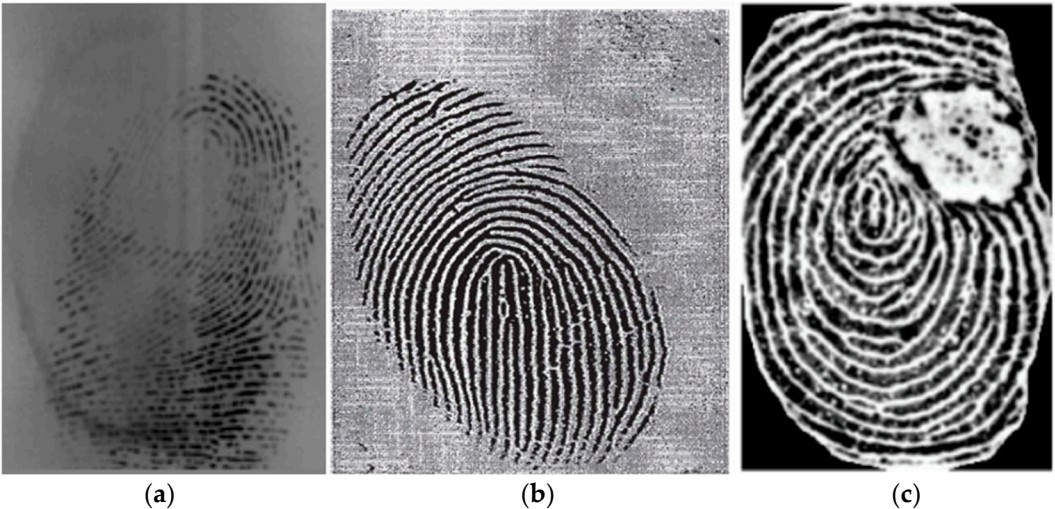

(**a**)                                     (**b**)                                     (**c**)

**Figure 1.** Different challenges for contact-based fingerprint images: (**a**) blurry images, (**b**) distorted image capture, (**c**) deformed images [21–23].

The contactless finger image obtains ridge–valley contrast that is different from a print made from the contact of a finger on a live-scan capture device [24]. To address this, different technologies for acquiring finger photos, such as 2D and 3D fingerprints, have been introduced [13,16,25,26]. Image processing can solve some issues, while the rest of them remain in contactless 2D and 3D fingerprint areas [27]. In recent years, deep-learning technology has demonstrated success in image recognition, classification, and feature representation [28–37]. These deep-learning models have also been employed in contactless fingerprint-based biometric technologies [38]. It is necessary to conduct a comprehensive survey on the latest research findings on 2D and 3D contactless fingerprint recognition systems based on deep-learning technology to understand those models and point out the future development direction. It is useful to note that there is a particular system of capturing contactless finger photo images that might impact the performance of the deep-learning models. Also, it will be useful to know how the limitations of classical machine-learning open the door for deep learning in the contactless fingerprint area. This study explored deep neural network (DNN) methods for contactless fingerprint recognition. For this, we needed to analyze the machine learning (ML)-based algorithms to compare with DNN methods. Photo capture and image processing are the first steps for contactless fingerprint recognition. We have explored the steps of feature extraction and recognition based on ML and DNN methods. Various test outputs with their performance were analyzed to validate the feasibility of the suggested DNN methods. This paper has investigated fingerprint capturing methods, fingerprint preprocessing, and feature extraction in both classical image processing and machine learning, as well as replacement of classical methods by deep learning. A total of 32 papers (without duplication) were found related to these topics. Following the application of inclusion and exclusion criteria, eight papers were selected for a full-text review.

The paper is organized as follows: first, the systematic review procedure is represented in Section 2 along with the description of three research questions (RQ). Section 3 presents a detailed investigation of the image-capturing method using image sensors. Section 4 explores relevant classical methods. Section 5 analyzes the deep neural network methods in contactless fingerprint recognition systems. Section 6 provides a discussion and Section 7 ends with conclusions for future work.

## 2. Review Methodology

The key focus of this review is an up-to-date summary of recent novel approaches. The systematic search procedure was set primarily following the Preferred Reporting Items for Systematic Reviews and Meta-Analyses (PRISMA) [23]. This methodology used the following processes: (a) identifying research question (RQ), (b) source of study (c) search strategy: setting inclusion/exclusion criteria, (d) results.

### 2.1. Research Questions

(1) RQ1. How do different sensor systems capture finger images to ensure the acceptable quality of fingerprints? Research findings will help to investigate whether the capturing systems have any impact on the model architecture or the recognition performance.
(2) RQ2. How does the classical machine-learning method preprocess the contactless finger images and prepare for recognition algorithms? Research findings explore the classical methods used for feature extraction, image segmentation, minutiae point extraction, image deblur, background noise removal, particular portion segmentation, and suitable feature extraction from finger images.
(3) RQ3. How do deep neural networks replace the classical recognition models? The answer will explore the architecture of related deep neural networks and their performance improvement over traditional methods.

### 2.2. Source of Studies

The search for relevant literature was performed across six repositories: Google Scholar, Science Direct, Wiley Online Library, ACM Digital library, MDPI, and IEEE. Search dates ranged from inception to 30 April 2022.

### 2.3. Search Strategy

The following 'free- text search terms' were used: 'finger photo recognition', 'fingerprint identification', 'touchless fingerprint recognition'. The search results were strictly restricted to the English language. References from selected primary full-text articles were further analyzed for relevant publications. The selection was further narrowed by applying the eligibility criteria described in Table 1. Articles fulfilling the inclusion criteria were considered in this review, and those fulfilling the exclusion criteria were filtered out.

**Table 1.** Inclusion and Exclusion Criteria for this systematic review.

| Inclusion Criteria | Exclusion Criteria |
| --- | --- |
| Article published in peer-reviewed venues | Papers not written in English |
| Article published since 2010 | Traditional contact-based fingerprint method |
| Articles must address a certain combination of words i.e., deep learning + contactless fingerprint recognition | |
| Automate + fingerprint identification, 3D + contactless identification, smartphone/mobile + capture, contactless + finger photo | |

A total of 49 publications were identified through the database search and three from the bibliography of those publications; however, 33 failed to satisfy the eligibility criteria

and were excluded. Thus, 16 publications ultimately fulfilled the eligibility criteria for this review. Figure 2 illustrates the methodology and results of the review process.

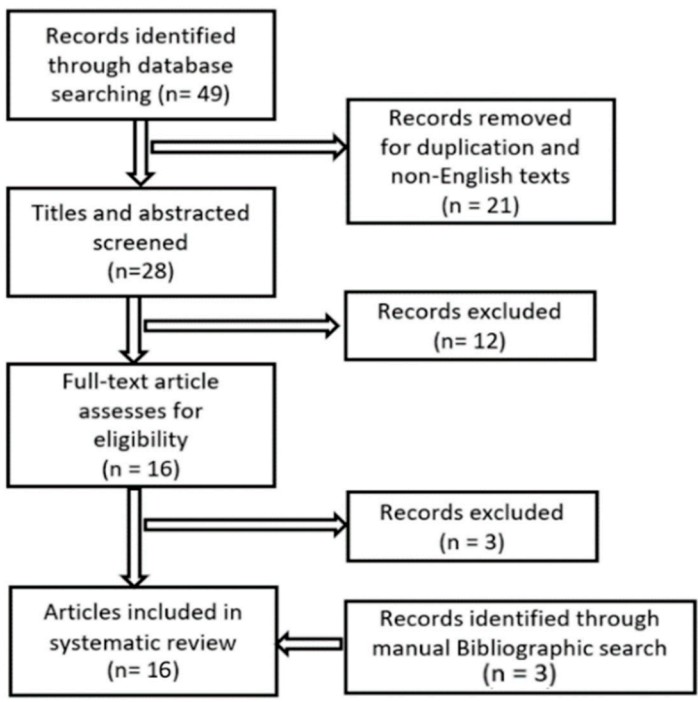

**Figure 2.** Flow diagram depicting the systematic review strategy.

*2.4. Review Outcome*

There are two types of contactless fingerprint capturing techniques: 2D and 3D. Smartphones and digital cameras can take 2D and 3D photos [17,39]; 3D contactless fingerprints can also be acquired with photometric stereo-based cameras [39], 3D fingerprint reconstruction, structured light-scanning-based 3D fingerprint reconstruction, etc. [40–47]. The general biometric workflow of a contactless fingerprint recognition system is described in Figure 3.

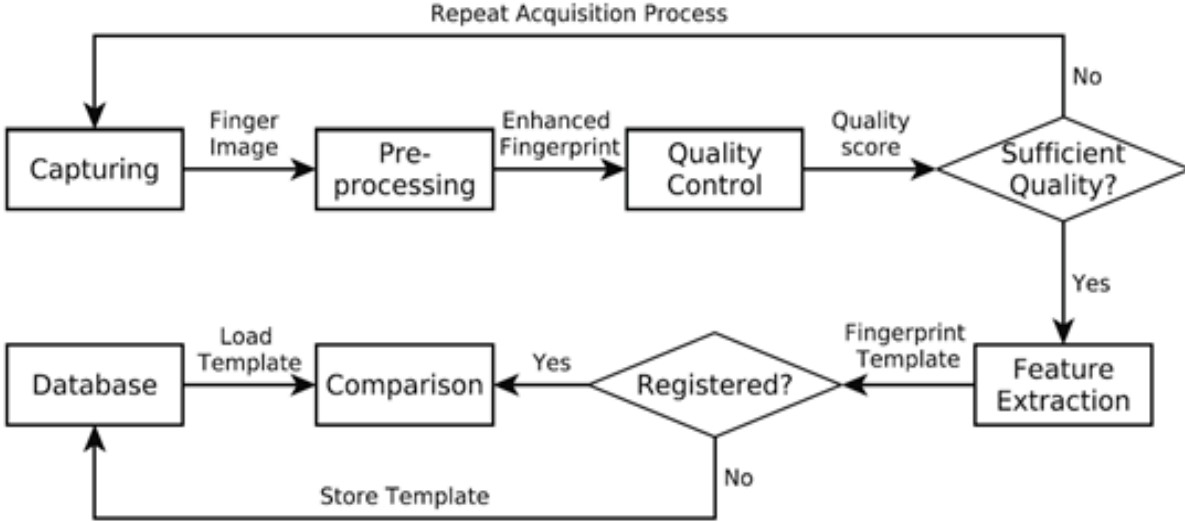

**Figure 3.** Overview of the sub-systems of a generic contactless fingerprint recognition system [10].

## 3. Contactless Fingerprint Capturing Methods

### 3.1. 2D Contactless Fingerprint Capturing Methods

During a contactless fingerprint capture, one or multiple fingers are presented to an optical device like a camera or lens. These devices can be (a) prototype hardware designs developed by researchers or (b) general-purpose devices customized to meet the unique needs for contactless fingerprint recognition [48].

Smartphone-based image acquisition is one of the widely available techniques to capture 2D contactless finger photos [4]. The NIST [17] published a document to assess contactless fingerprint capturing methods; the document provides proper instructions for contactless fingerprint capturing devices. It describes the smartphone's uniform light lighting, backdrop segmentation, and motion reduction during capture. Multi-finger capturing techniques can also be used with smartphones [49]. The advantage of multi-finger capture is the efficiency since feature extraction of all five fingers can occur from one single image [50]. To evaluate the performance of Nokia N95 and HTC Desire mobile phone for this, 1320 fingerprints were captured. A flashlight-enabled phone was used for appropriate illumination and to cover the entire finger area. However, the image quality was reported to be poor, as a flashlight performs well only in a dark setting [22]. To improve image quality and to reduce camera noise, dark environments might play a very important role. Auto-focus and maintenance of a standard distance from hand to phone may also be useful strategies. Figure 4 shows the identical distance and illumination from hand to phone.

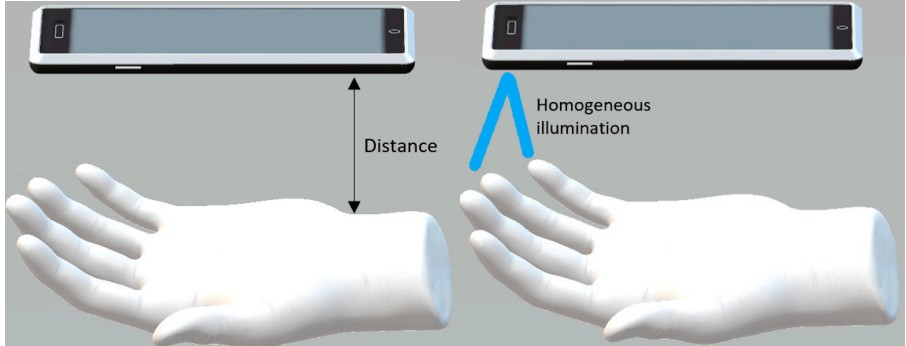

**Figure 4.** Homogeneous distance illumination with auto focus capturing by smartphone.

Figure 5 illustrates the impressions of a fingerprint taken with a contact-based fingerprint device (Figure 5a) and the corresponding finger image acquired using a non-contact device (Galaxy S8). (Figure 5b). The contact-based fingerprint can directly be used for finding feature such as: ridge, valley, delta cores, minutiae etc., whereas the corresponding contactless fingerprint image would need additional processing.

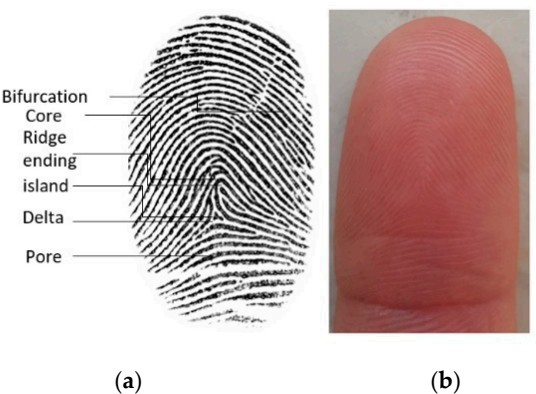

(**a**) (**b**)

**Figure 5.** (**a**) Contact-based fingerprint, (**b**) contactless fingerprint (Samsung Galaxy S8) [15].

A digital camera is another tool to capture contactless 2D finger photos. The main feature of this system is white-color- and LED-color-based image sensors [10]. A three-camera-based system with blue LED is much more comfortable than a white LED for the user to acquire fingerprints. The charge-coupled camera device emits a green LED, and there is a stepper motor with a mirror that can capture five fingers at a time, making it convenient [11].

### 3.2. 3D Contactless Fingerprint Capturing Methods

Researchers have used lab-developed prototypes of the 3D fingerprint capture approach, which requires (i) photometric stereo techniques, (ii) structured light scanning, and (iii) stereo vision [51]. The photometric stereo-based 3D fingerprint method captures multiple 2D images under heterogeneous illumination using a high-speed camera. Time-of-flight (ToF) is the main principle of this technique, where surface reflectance is measured from fingerprint to light source [52]. This system is low-cost because it uses only one high-speed camera and multiple LEDs. Figure 6 shows the capturing method with camera and finger position.

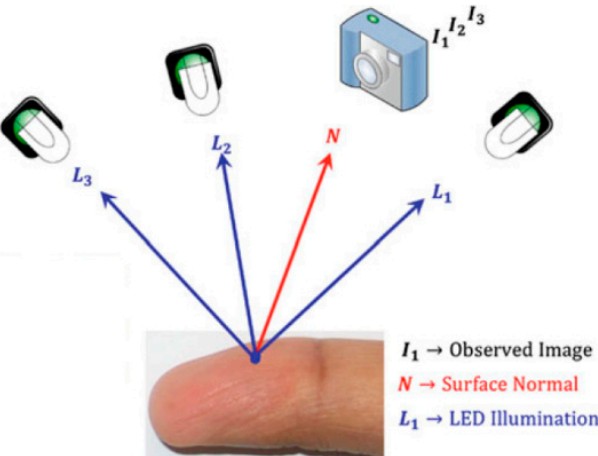

**Figure 6.** Acquisition of 3D fingerprint using photometric stereo techniques [39].

The structured light-scanning method consists of several high-speed cameras and a digital light-processing projector [53,54]. During the capture process, multiple 2D fingerprint images are captured under pattern illuminations, and 3D depth information is calculated through triangulation according to the point correspondences between images [55]. This method can recover ridge-valley details and achieve relatively accurate 3D depth information. However, the hardware system is expensive and bulky [39].

The stereo-vision-based 3D contactless fingerprint method is usually comprised of two or more cameras [51,56,57]. During the capture process, 2D fingerprint images are captured from different views. The 3D fingerprints are reconstructed by calculating 3D depth information between corresponding points according to the triangulation principle. The advantage is that the systems are simple, low-cost, and relatively compact. However, current methods are usually time-consuming because of the extensive computation of the correspondences between pixel points [58]. Table 2 shows the 2D and 3D capturing devices and their approximate cost, with light environment, etc.

**Table 2.** Overview of contactless 2D and 3D capturing devices and their properties.

| Capturing Device | Authors | Cost | Light Environment | Finger Type |
|---|---|---|---|---|
| **Mobile Phone (2D)** | Lee et al. [19] | Low cost | No extra illumination | Single Finger |
| **Digital Camera (2D)** | Hiew et al. [59] | Low cost | Table lamp illumination | Single Finger |
| **Digital Camera (2D)** | Genovese et al. [60] | Medium cost | Green Light illumination | Finger slap |
| **Webcam (2D)** | Piuri et al. [61] | Low cost | Different illumination (white light, no light) | Single Finger |
| **Webcam** | Kumar and Zhou [9] | Low cost | No illumination | Finger slap |
| **Smartphone (2D)** | Derawi et al. [18] | Low cost | No illumination | Finger slap |
| **Smartphone (2D)** | Canrey et al. [49] | Low cost | Screen guidance. If flash required (Y/N) | Finger slap |
| **Smartphone (2D)** | Deb et al. [61] | Medium cost | 3 smartphones in different illumination | Thumb and index finger |
| **Smartphone (3D)** | Xie et al. [51] | Medium cost | 2 cameras with depth information | Finger slap |

## 4. Classical Method to Extract Features from Contactless Fingerprints

The basic steps for contactless fingerprint recognition pipeline are shown in Figure 7 as a flowchart:

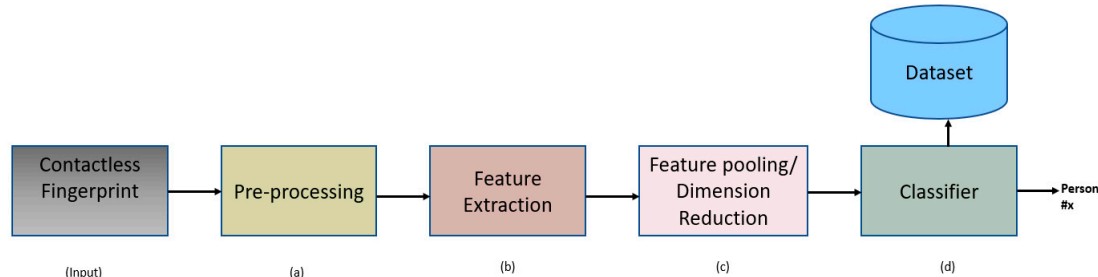

**Figure 7.** The fundamental steps for contactless fingerprint recognition from input to (**a**) preprocess the images; (**b**) Feature extraction (**c**) Dimension reduction; and (**d**) classified the person.

Most of the contact-based fingerprint images captured from the devices are grayscale and ready for feature extraction [62]. In contrast, most contactless finger-imaging solutions provide color RGB images that require preprocessing before feature extraction [63]. The primary challenges of preprocessing contactless finger images and recommended methods to overcome these challenges are shown in Table 3.

**Table 3.** Overview of challenges during the preprocessing of contactless finger images and proposed approaches.

| Challenge | Authors | Year | Approach |
|---|---|---|---|
| Finger Segmentation | Wang et al. [62] | 2017 | Hand color estimation in YCbCr |
| Rotated pitched principal orientation estimation | Zaghetto et al. [9] | 2015 | Artificial neural network |
| Low contrast | Wang et al. [62] | 2016 | CLAHE and extensions |
| Distance to the sensor, ridge line frequency | Zaghetto et al [9] | 2017 | Frequency map, sensor-finger distance approximation |
| Core/principal singular point detection | Labati et al. [64] | 2010 | Poincare-based ridge orientation analysis |
| Deformation correction | Lin et al. [11] | 2018 | Robust thin-plate splines, deformation correction model |

When processed with the classical methods, both contactless 2D and 3D images have issues with low focus of ridge/valley and blurred ROI (background) [57]. Misplaced or rotated fingers and the lack of skin deformation also cause processing issues [65]. An image processing pipeline must be developed based on the selected equipment and the environmental conditions needed during the image acquisition process. Image processing begins with the following common preprocessing steps:

Finger Segmentation and Detection: The initial step is to detect the finger based on color and shape. Sharpness, shape, color contrast, and image depth information are four different categories for improving contactless 2D and 3D finger detection and image segmentation [13]. Sharpness-based strategies utilize the difference between the focused, blurred background and the sharp finger area. This effect works best with images obtained with a very small finger-to-sensor distance and a wide-open aperture. One experiment showed that the variance-modified Laplacian of Gaussian (VMLOG) algorithm is best suited for contactless 2D fingerprint-capturing devices [10,66,67]. All finger shapes, from thumb to little finger, are highly similar for all finger position codes. A machine-learning-based algorithm has been applied to a binarized image in the LUV color model [10]. To make the skin color contrast and segment the skin and background color, the analysis of the YCbCr color space represents a very vital approach [64,68]. A different method of image segmentation and image depth information approach combined an RGB image via a smartphone [69]. These were able to extract the finger slap (the four fingers except the thumb) from busy backgrounds for further processing.

Minutiae-Based Feature Extraction: One of the main conditions for pre-processed contactless fingerprints is that images must be converted from RGB to greyscale [70]. Thus, ROI such as minutiae, ridge valley extraction, and finger orientation estimation must also be handled with a machine-learning approach. After detecting the finger, the ROI must be extracted, which involves the normalization of width, height, and resolution. This 3D contactless fingerprint preprocessing stage implies an extracted finger image as input. It should be noted that finger detection and ROI extraction are done in output. The color-based segmentation of ROI extraction constrained setups depends on contactless 3D finger geometry [10]. Several operations used the ridge–line orientation and shape to detect the core point [71]. Using a support vector machine (SVM) [72], it is easy to classify minutiae-based fingerprints and to detect the minutiae points as the detection points to refer to as a category [73]. SVM can determine the image quality with five feature vector lengths such as gray mean, gray variance, contrast, coherence, and the main energy ratio. These features take much training time to implement.

Fingerprint Image Enhancement: To improve image contrast and sharpness, image enhancement techniques such as spatial domain techniques and frequency domain techniques can be used to improve the quality [74,75]. Finger image enhancement should result in a fingerprint image with uniform illumination. Three different methods to achieve this appeared in the literature: a normalization using mean and variance filters [30], histogram enhancements like contrast-limited adaptive histogram equalization (CLAHE) [76], and local binary patterns (LBP) for enhancing the ridge–valley contrast of the 2D contactless fingerprint system [77]. Reducing the blurred image from the original image is another challenge in contactless 3D fingerprint enhancement [14]. A combination of image-processing algorithms and machine learning for extracting sweat pores of fingerprint patterns level-3 has been proposed by Genovese et al. [78].

## 5. Analyzing the Deep Neural Networks Methods Proposed for the Contactless Fingerprint Recognition Systems

Simple convolutional and pooling layers were utilized to create deep-learning models in many articles, but a multi-task fully convolutional network was used in three of them. The architecture for the multi-task deep convolutional network is shown in Figure 8.

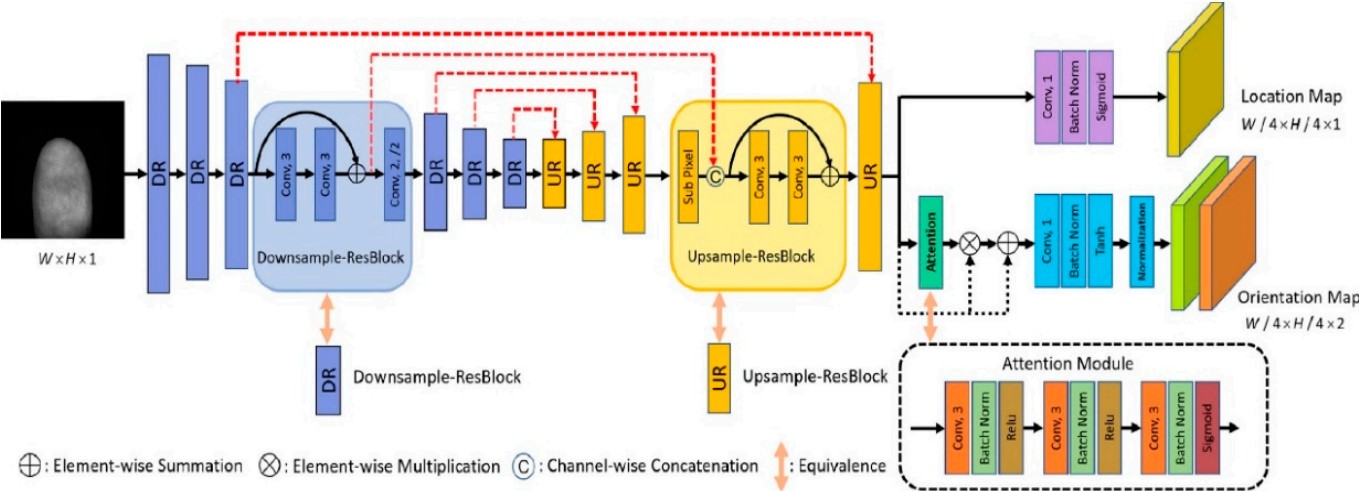

**Figure 8.** An architecture of multi-task deep convolutional networks [79].

Matching the contactless fingerprint with a traditional contact-based fingerprint using deep learning is a new domain in biometrics research. In order to recognize contactless fingerprints, this paper [67] described a convolutional neural network (CNN) framework. The convolutional and pooling layers are the two main layers of the algorithm. The convolutional layers were used to execute low-level features such as edges, corners, etc. Pooling layers enabled correct operations such as reducing the dimension of feature maps. Ten images were provided to the CNN model as an input batch for training. A training accuracy of 100 percent was attained after four iterations. At 95 percent of testing accuracy, 140 out of 275 images were used for testing purpose.

A fully convolutional network was applied for minutiae detection and extraction in [79]. The minute point and its corresponding direction were processed and analyzed using contactless grayscale fingerprint images from two different public datasets [12,80]. Images were assessed online after being trained offline. A full-sized contactless fingerprint from two different datasets (9000, 6000) was applied as an input and its corresponding minute ground truth was indicated as an output in the offline portion. In conjunction with a novel loss function, this method concurrently learns the minutiae detection and orientation. One of the main claims of this study is that a multi-task technique outperforms any single minutiae detection task. An hourglass-shaped encoder–decoder network [81] structure was applied for a multi-task deep neural network called ContactlessMinuNet architecture [79]. To process the input fingerprint images, a shared encoder subnetwork was used. For up-sampling, the subnetwork was decoded to expand the image. Lastly, the network split into two branches for minutiae detection and direction computation.

Minutiae point detection branch: In this network [79], the input feature represents the detection of minutiae points, and the output represents the probability of each pixel's minutiae points. The network is very simple, with a $1 \times 1$ convolutional layer, a batch normalization layer that standardizes the input layer, and a sigmoid layer. A non-linear activation function sigmoid layer is used to generate minutiae location.

Minutiae direction regression branch: This network [79] is designed to predict the minutiae direction. Pixel-by-pixel images were extracted with a phase angle $\varnothing i \in (0,2\pi)$. The subnetwork works as input features, and the output layer predicts minutiae direction. A convolutional layer ($1 \times 1$), batch normalization, and non-linear activation function *tanh* layer were used to predict the minutiae path.

Using $3 \times 3$ CNN layers, the final convolution has been used with (stride = 1) and padding to keep the height and width constant. For testing, the proposed method was compared with the benchmark dataset of the PolyU dataset [12]. The accuracy of minutiae detection and its location increased to 94.10% compared to 89.61%. The proposed method of this study is shown in Figure 9.

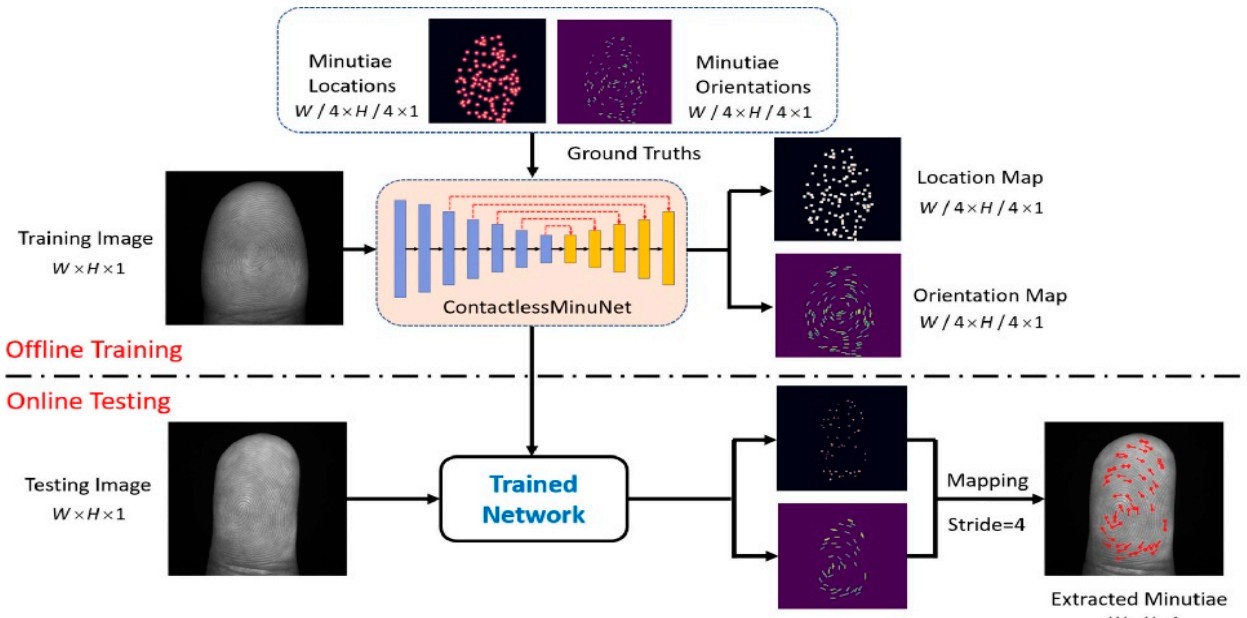

**Figure 9.** Overview of the minutiae extraction algorithm for contactless fingerprints based on multi-task fully deep convolutional neural network [2].

A study reported in [82] suggested how to extract a minutiae point from an input image without preprocessing. To train the model and obtain the output without any preprocessing, a number of deep neural networks have been deployed. Initially, JudgeNet was trained to locate the minutiae regions and picked a general overview of detecting minutiae points. The original image resolution was $640 \times 640$ and 500 ppi with 200 labeled images. A max pooling was used to reduce the image dimensions, and it showed image dimensions of $45 \times 65$. Using multiscale input and four CNN layers, the network performed very well to get the accurate output. Later, another deep CNN layer named LocalNet specifically indicated the directions of minutiae with a more concise image dimension of $45 \times 45$ and decided the specific location. Lastly, a comprehensive estimation and decision were made to add or eliminate the minutiae location. The overview of the network architecture is shown in Figure 10.

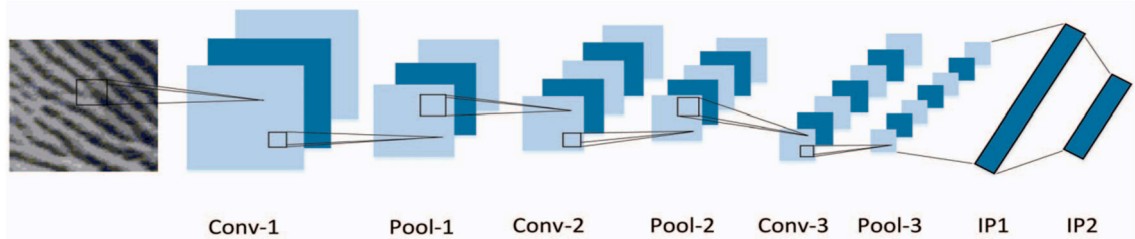

**Figure 10.** JudgeNet and LocalNet share a similar convolutional architecture [82].

A method was proposed [83] to get the proper position of the contactless fingerprint of multi-view 3D fingerprint features using CNN. A fully convolutional network (FCN) was applied with this model for automatic fingerprint segmentation and three Siamese networks for fingerprint multi-view. Various convolutional neural network models such as VGG net [31], AlexNet [84], and GoogleNet [85] are introduced in this work. These architectures are very well-trained pixel-to-pixel deep networks. For foreground and background segmentation, semantic segmentation architecture was applied. This model clusters the same image together with a different class. To predict each pixel from the top-view of the fingerprint, they used the softmax loss function. The Siamese convolutional

network worked very well to match image pairs (matched and unmatched) in the same network. Three Siamese networks indicated the positions: top view, side 1 view, side 2 view. The network is structured with six convolutional layers with one fully connected layer. One to five layers are followed by max-pooling, where the input patch size was 256 × 192.

The kernel size was 3 × 3 with stride value 2. The output numbers from the feature map were 64, 96, 128, 256, and 512, generated from the 48-feature map in the first convolutional layer. The final result was presented in the receiving operating characteristic (ROC) curve and the equal error rate (EER) curve to evaluate performance. Using computing matching scores from CNN-based features and minutiae-based features, another Siamese convolutional neural network was applied to extract the global feature from a finger photo. They mentioned that fingerprint images have many global features that ease extraction of the features using CNN. Using an input image size of 310 × 240, the first convolutional layer was introduced with a kernel size of 3 × 3 with batch normalization. By evaluating the ROC curve and EER curve, they showed the estimation of EER; minutiae matching rate was 11.39% and 4.09%, respectively. Figure 11 shows how a fully convolutional network segments the fingerprint background and directs the capturing methods of multi-view with deep representation:

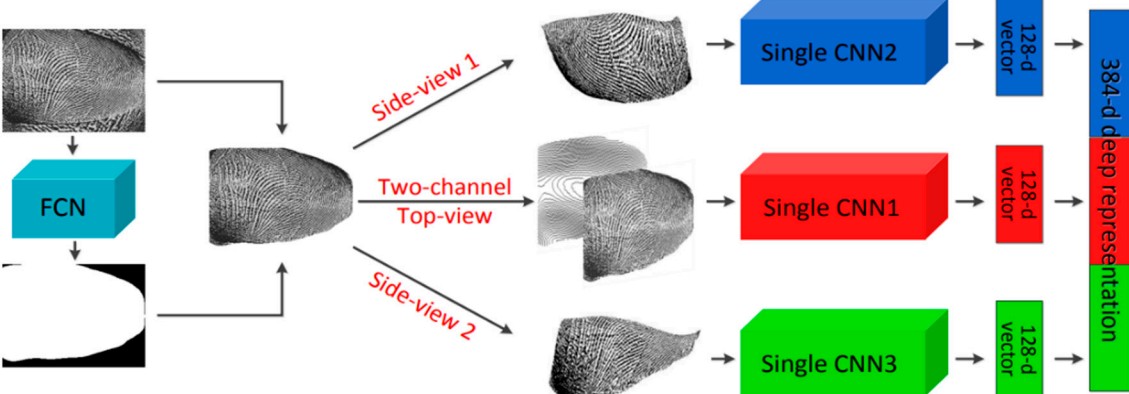

**Figure 11.** Automatic learning of 3D fingerprint features via deep representation [86].

A CNN-based framework has been applied [78] to make an accurate comparison between contactless fingerprints and contact-based fingerprints. Minutiae points, ridge maps, and specific regions are the targeted metrics to establish the comparison [83]. A multi-Siamese network was used to train and learn the minutiae features. As the image dataset was collected from different sensors, a CNN-based cross-comparison framework was used to compare contactless and contact-based fingerprints. Figure 12 shows the deep-feature representation generation process:

A data augmentation process was used in 5780 contactless and contact-based images from 320 fingers [78]; 3840 images were used in the training set while the rest were used for testing purposes. The image size remained 192 × 192 in every image. A public dataset was used to determine the performance and to compare and validate the dataset. The dataset contains 1500 fingers data with 3000 contactless fingerprint samples. For the performance metrics and evaluation, the ROC method (receiver operating characteristics) and EER (equal error rate) were used. To obtain the fingerprint recognition, CMC (cumulative match characteristics) and rank-one accuracy methods were applied. The comparative experimental evaluations are shown in Table 4. Also, in Table 5 shows the total summary of the described articles of deep learning in Section 5.

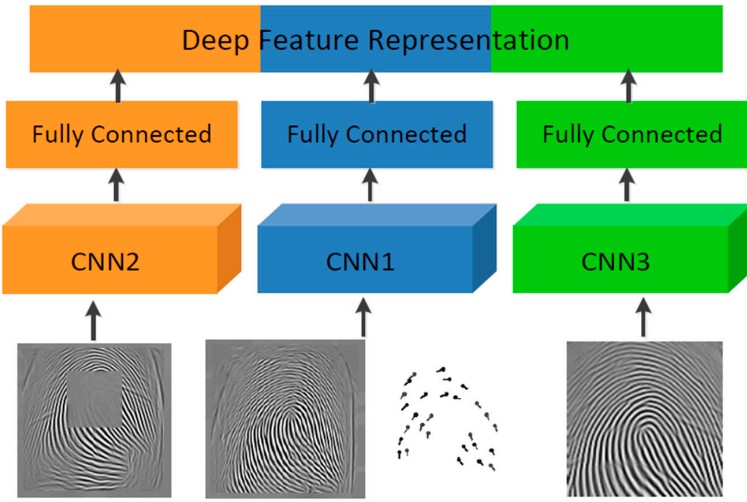

**Figure 12.** Deep-feature representation generation process using three multi-Siamese networks [83].

**Table 4.** Experimental evaluation of two datasets.

| Experiments | Equal Error Rate (ERR) | Rank-One Accuracy |
|---|---|---|
| Deformation correction model [87] on dataset A | 16.17% | 41.82% |
| Minutiae matcher in NIST [88] on dataset A | 43.83% | 10.99% |
| Proposed method on dataset A | 7.93% | 64.59% |
| Deformation correction model [87] on dataset B | 21.60% | 38.90% |
| Minutiae matcher in NIST [88] on dataset B | 38.01% | 24.92% |
| Proposed method on dataset B | 7.11% | 58.87% |

**Table 5.** The following table describes the summary of the analysis.

| Study | Database | Training Data | Purpose of Deep Learning | Input to Deep Neural Network | Output from Deep Neural Network | Performance Metrics |
|---|---|---|---|---|---|---|
| [67] | Private | 275 images with 55 different people | Fingerprint Recognition | RGB to Gray scale images | Feature matching | Classification (Metric Accuracy) |
| [78] | Public | 5760 images from 320 fingers | Minutiae Extraction | Gray scale images | Extracted minutiae images | AUC, EER |
| [79] | Private + Public | 9000/6000/ 1320 images | Multiview fingerprint recognition | Gray scale images | Feature (Ridge, valley) representation | EER |
| [80] | Public | 100 images | Minutiae Extraction | Gray scale images | Extracted Minutiae images | Classification (Metric Accuracy) |
| [82] | Private + Public | 500 images | CNN based framework for Contactless fingerprint | HSV images | Similarity distance between two images | ROC curve |
| [83] | Public | 9920 images | To correct fingerprint viewpoint | Gray scale images | Correct images | ROC and CMC curve |

## 6. Discussion

This review is intended to provide a systematic survey of the deep-learning approaches employed for contactless fingerprint processing. The review surveyed eight full-text scientific research articles showing how deep learning can replace machine learning in contactless fingerprint contexts. This review was focused on three major research questions: the contactless finger photo capturing method, the classical approach of fingerprint recognition, and the use of deep learning. The first research question shows the direction of the different capturing methods and various camera sensors. This analysis might help researchers to understand divergent capturing methods and their limitations. Also, they might be motivated to employ smartphone-based finger photo capturing. In the second research question, we explained how classical machine-learning techniques have been introduced into contactless fingerprint recognition methods. This review covered feature extraction, image segmentation, and blur reduction; however, systems like data acquisition, data cleaning, data labeling, etc. were beyond the scope of this review.

The main contribution of this paper is the review of the use of deep learning, specifically, its impact and usability in the field of contactless fingerprints. We have discussed how a contactless recognition system can benefit from using deep learning. In addition, we have also pointed out potential vulnerabilities in classical methods and shown the applicability of deep learning to real-world applications. Research Question 3 shows that the following factors can impact the research of contactless fingerprint recognition systems:

Feature learning: Deep-learning methods have an advantage over previous state-of-the-art methods because they can learn features from data. Contactless fingerprint recognition systems require both local and global features [89] and are compatible with hierarchical and structural feature learning enabled by deep learning [90]. In addition to processing and labeling with the handcrafting data, some tools like labelme and image-label will be difficult in most cases. Therefore, deep learning can assist in preprocessing or extraction of the features of fingerprint images. The learned features can be generalized to previously unseen datasets and other related tasks (for example, features learned for contactless fingerprint recognition can also be used for fingerprint attribute estimation, e.g., ridge, minutiae pattern). In addition, pre-training improves feature-learning by large amounts of unlabeled data when using smaller training datasets.

Concentration on Identification: Authentication and recognition have been the primary focus of deep-learning research in the contactless fingerprint context. Authentication is a comparably easy problem and estimates well for a large number of subjects. However, the more challenging part is the identification problem. The biometric system needs to distinguish between potentially millions of identities for large-scale identification. This system requires complex deep-learning architectures to capture definite interclass differences and handle large intra-class variability. Consequently, much training data would be required to capture these variations.

Large-scale datasets: Though deep-learning approaches have already exceeded human performance on some in-the-wild, large-scale datasets, these datasets do not meet the requirements of real-world, high-security applications. In addition, there is a lack of large-scale datasets for contactless fingerprint modalities in biometrics to benefit from deep learning. Even if large datasets are available, each individual needs to have sufficient representative samples to consider for various influencing factors.

Dataset quality: Existing fingerprint recognition datasets are mostly gathered from the public dataset. It is important to use large-scale datasets that capture real-world variations for biometrics to benefit from deep learning, especially in the contactless fingerprint area.

Computing resources: Along with the increased use of mobile devices, secure authentication commercial devices have become necessary modern technologies. However, if complex deep-learning architectures are required for authentication, such devices might not have the necessary computing resources for storing the dataset. A cloud-based system could be a solution for restoring the data collected from those devices.

Training deep-learning models with proper computing resources: The success of deep learning has been largely demonstrated by industries with access to large amounts of data and computational resources. For most other researchers, computing resources are limited, and it is imperative to speed up the training of deep-learning approaches. We need to strive for data-efficient learning algorithms.

## 7. Conclusions

This review focused on three major research questions: the contactless finger photo capturing method, the classical approach to fingerprint recognition, and the use of deep learning. Specifically, we have detailed deep-learning methods, as these methods have shown development in contactless fingerprint recognition, though little has been explored. The accuracy of contactless fingerprints is increasing day by day and they have facilitated a new range of fingerprinting applications. They have increased the security system threats with respect to terrorism and cyber-crime development. Commercial facilities, border crossing areas, airports, and government access points are also employing contactless fingerprint biometrics. Further, credit card account fraud, hijacking of websites, and most importantly, the critical corruption of governmental agencies such as the Department of Defense and the Department of Homeland Security require the development of systems for which contactless fingerprint biometrics can be a solution. These deep-learning methods have demonstrated good generalization capability for different datasets. We have summarized their architecture and implementation at various sub-stages, including pre-processing, features extraction, classification, or matching. This study also covered the possible drawbacks of deep-learning models.

In summary, deep-learning-based contactless 3D fingerprint identification systems have shown enhanced usability, and soon they will be a widely used biometric performance modality. Therefore, our future research will be focused on creating new or existing deep-learning techniques to address certain upcoming contactless fingerprint challenges, such as speeding up feature extraction, reducing the amount of time required to process images, and improving identification accuracy. Additionally, other biometric characteristics such as patterns in palmprints will be taken into consideration as applications of deep-learning techniques.

**Author Contributions:** A.M.M.C. prepared the article and M.H.I. Supervised him. All authors have read and agreed to the published version of the manuscript.

**Funding:** This research received no external funding.

**Institutional Review Board Statement:** Not applicable.

**Informed Consent Statement:** Not applicable.

**Data Availability Statement:** Not applicable.

**Conflicts of Interest:** The authors declare that no conflict of interest.

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
