# Peer review of "Contactless Fingerprint Recognition Using Deep Learning—A Systematic Review"

_jcp, doi:10.3390/jcp2030036_

Round 1

Reviewer 1 Report

In this paper, authors studied contactless fingerprint recognition using deep learning. Various research studies are reviewed through finger photo capture scheme, classical preprocessing scheme, deep learning approach. Basically, the motivation of the paper is fair and the covered topics and contents fit with this periodical. Also, as a whole, the organization and the presented quality of the paper are both good. It follows that readers can approach the manuscript without critical misunderstanding.

To improve the paper, the below minor parts are recommended to be addressed by authors carefully.

-If the space is available, authors can investigate more critical related works and add those studies to the manuscript.

-In the manuscript such as Conclusion, authors can describe future research challenges and issues for potential researchers.

-In Section V, authors need to provide more detailed information of experiment environment including system settings, used tools, hardware, the brief reason why they utilized those settings, etc.

-Authors can check if there are minor typos and unclear sentences in the manuscript.

-It is also recommended that authors will provide better visibility of some figures. For example, the font size of Figure 8 can be increased for better visibility.

Author Response

We would like to thank the reviewers for their careful review of our paper and their useful comments and suggestions to improve the quality and presentation of the paper. We have thus accordingly modified the paper. The following is our response to the comments of all reviewers individually. We hope that the revision is satisfactory, and the paper would now be found suitable for the publication.

Reviewer 2 Report

The topic Contactless Fingerprint Recognition Using Deep Learning- A Systematic Review is potentially interesting, however, there are some issues that should be addressed by the authors: The Introduction" sections can be made much more impressive by highlighting your contributions. The contribution of the study should be explained simply and clearly. The authors should further enlarge the Introduction with current work about recent machine learning algorithms  to improve the analysis of the system performance and the research background, for example: Reliable Deep Learning and IoT-Based Monitoring System for Secure Computer Numerical Control Machines Against Cyber-Attacks With Experimental Verification; Development of an IoT Architecture Based on a Deep Neural Network against Cyber Attacks for Automated Guided Vehicles; a systematic review of finger vein recognition techniques‏‏.

Clarify how the parameters of the machine learning techniques are adjusted

Clarify how the authors handle the overfitting issue related to the datasets

Clarify the practical implementation according to the cost

Conclusion section should be rearranged. According to the topic of the paper, the authors may propose some interesting problems as future work in the conclusion.

This study may be proposed for publication if it is addressed in the specified problems.

Author Response

(The authors gave the same response as above.)

Reviewer 3 Report

Please view the attachment

Author Response

(The authors gave the same response as above.)

Round 2

Reviewer 2 Report

Accept

Author Response

Thank you so much for your valuable comments.

Reviewer 3 Report

Thank you for addressing all the comments. This version looks better now. The section label could still improved by using A,B,C, 1,2,3 instead of a mixture of ii. and ii). The numbering of "i)" on line 194 is still a bit strange.

Author Response

Thank you so much for your valuable comments.

We have updated the manuscript by addressing the sections as "1, 2, 3," and under the sections the sub-sections are as "i", "ii" etc.